# Biomaterials for Drugs Nose–Brain Transport: A New Therapeutic Approach for Neurological Diseases

**DOI:** 10.3390/ma14071802

**Published:** 2021-04-06

**Authors:** Roberta Cassano, Camilla Servidio, Sonia Trombino

**Affiliations:** Department of Pharmacy, Health and Nutritional Sciences, University of Calabria, Arcavacata di Rende, 87036 Cosenza, Italy; roberta.cassano@unical.it (R.C.); camillaservidio@gmail.com (C.S.)

**Keywords:** stimuli-responsive hydrogel, intranasal administration, nose to brain, neurodegenerative diseases, Alzheimer’s disease, Parkinson’s disease, drug delivery

## Abstract

In the last years, neurological diseases have resulted in a global health issue, representing the first cause of disability worldwide. Current therapeutic approaches against neurological disorders include oral, topical, or intravenous administration of drugs and more invasive techniques such as surgery and brain implants. Unfortunately, at present, there are no fully effective treatments against neurodegenerative diseases, because they are not associated with a regeneration of the neural tissue but rather act on slowing the neurodegenerative process. The main limitation of central nervous system therapeutics is related to their delivery to the nervous system in therapeutic quantities due to the presence of the blood–brain barrier. In this regard, recently, the intranasal route has emerged as a promising administration site for central nervous system therapeutics since it provides a direct connection to the central nervous system, avoiding the passage through the blood–brain barrier, consequently increasing drug cerebral bioavailability. This review provides an overview of the nose-to-brain route: first, we summarize the anatomy of this route, focusing on the neural mechanisms responsible for the delivery of central nervous system therapeutics to the brain, and then we discuss the recent advances made on the design of intranasal drug delivery systems of central nervous system therapeutics to the brain, focusing in particular on stimuli-responsive hydrogels.

## 1. Introduction

Neurological disorders represent the first cause of disability and the second cause of death, around 17%, after cardiovascular diseases. Unfortunately, in the following years, the number of patients suffering from neurological disease is expected to rise dramatically due to the increase in average life expectancy, thus causing a heavy global burden [1,2,3].

The most frequent neurologic diseases are Alzheimer’s disease, Parkinson’s disease, multiple sclerosis, dementia, epilepsy, schizophrenia, stroke, and brain cancer. Although neurological disorders present different clinical manifestations and pathogenesis, they all mainly determine a progressive neuronal degeneration. Their etiology is complex and not completely known; however, genetic and environmental factors and aging are considered to have a key role [4,5,6,7].

Current therapeutic approaches against neurological disorders include oral, topical, or intravenous administration of drugs and more invasive techniques such as surgery, brain implants (i.e., deep brain stimulations), and so on. However, unfortunately, the currently available treatments are not associated with a repair and/or regeneration of the neural tissue and, consequently, resolution of the clinical situation but primarily act on alleviating symptoms and slowing the neurodegenerative process [8,9].

The main limitation of central nervous system therapeutics is related to their delivery to the nervous system in therapeutic quantities. In this regard, the blood–brain barrier (BBB) plays a key role in preventing the passage of many substances, including drugs, to the brain [10,11].

The BBB is a complex vascular system consisting of endothelial cells interconnected by extended tight junctions without fenestrations, whose main function is to maintain the nervous system homeostasis. The endothelial cells are surrounded by pericytes and astrocytes that contribute to the structural and functional preservation of the BBB.

The BBB is the main structure responsible for the protection of the central nervous system from circulating xenobiotics, which could have harmful effects on the neural function, and of the supplying of required nutrients to the neural tissue [12,13].

The main pathways for crossing the BBB are passive diffusion for small lipophilic compounds, active or passive transport for specific hydrophilic and/or ionized molecules (i.e., glucose, amino acids), and, lastly, for high-molecular-weight biological molecules, such as proteins and peptides, endocytosis [14,15]. The high selectivity of these crossing pathways massively prevents the entrance of circulating xenobiotics into the central nervous system, and for this reason, it has been estimated that around 98% of drugs is not able to cross the BBB [16].

In order to overcome these limitations, the intranasal route has recently emerged as a promising administration site for central nervous system therapeutics [17,18]. In fact, it is a noninvasive route that provides a direct connection to the central nervous system, nose-to-brain route, via neural pathways, such as the olfactory and trigeminal ones, avoiding the passage through the BBB and, consequently, increasing drug concentration at the brain level [19,20] (Figure 1).

Moreover, the intranasal route shows also the advantage, compared to the oral route, to prevent the gastrointestinal and first pass degradation, thus improving drug bioavailability [21].

This review provides an overview of the nose-to-brain route: first, we summarize the anatomy of this route focusing on the neural mechanisms responsible for the delivery of central nervous system therapeutics to the brain, and then we discuss the recent advances made on the design of intranasal drug delivery systems of central nervous system therapeutics to the brain, focusing in particular on stimuli-responsive hydrogels.

## 2. Nasal Cavity: Anatomy and Physiology

The nose is the organ responsible for respiration and olfaction. Structurally, it is divided into the external nose and internal nose (nasal cavities).

The external nose is formed by bones and cartilages; it is placed in the center of the face and has the shape of a triangular pyramid. Small muscle groups controlled by the facial nerve that contribute to facial expression are connected to the external nose [22].

The nasal cavity extends around 12 cm in length from the external nose to the nasopharynx, and it is separated in two sections (left and right) by the nasal septum.

Structurally and functionally the nasal cavity can be divided into three regions: vestibular, olfactory, and respiratory [23].

The vestibular region is the smallest and outermost part of the nasal cavity lined firstly by stratified squamous epithelium followed by respiratory epithelium (pseudotratified columnar epithelium). It contains nasal hairs, called vibrissae, whose main function is to filter inhaled airborne particles, and sweat and sebaceous glands. This region can be considered irrelevant for drug absorption due to its structural features [7,24].

The respiratory region has the largest surface area and represents around 80% of the nasal cavity; in fact, it covers three turbinates, bones that extend laterally from each nasal cavity [25].

It is lined by pseudostratified columnar epithelium and contains four cell types: ciliated, nonciliated, basal, and goblet cells.

The respiratory region is covered by a thick layer of mucus produced by nasal glands and basal cells that plays a key role in trapping inhaled particles, preventing their entrance into the respiratory system. Ciliated cells contribute to this defense mechanism, called mucociliary clearance, transporting mucus to the nasopharynx where it is expectorated or ingested [26].

This region is highly vascularized, since it is supplied by a branch of the maxillary artery and it is mainly innervated by the trigeminal nerve [27].

The nasal region can be considered the main site for systemic drug absorption due to its large surface area, high vascularization, and viscosity and can contribute to the nose-to-brain delivery of drugs via the trigeminal pathway [28].

The olfactory region is located in the superior part of the nasal cavity (which represents about 10% of the nasal cavity area), and it is responsible for the olfaction. It is lined by pseudostratified columnar epithelium and contains four different cell types: basal, sustentacular, trigeminal, and olfactory neural cells [29].

Sustentacular cells are prevalent in the olfactory region and surround the olfactory neural cells, providing structural and metabolic support [27]. Olfactory neural cells are bipolar neurons that extend their dendritic processes into the mucus layer, terminating as olfactory receptors, and project into the olfactory bulb, providing a direct portal between the nose and the central nervous system. In particular, their unmyelinated axons are covered by olfactory ensheathing cells and olfactory nerve fibroblasts that are in continuity with meninges and, consequently, with the subarachnoid space [30].

The basal lamina is located below the epithelium and contains blood and lymphatic vessels, nerve fibers, and Bowman’s glands responsible for the secretion of mucus that, in turn, solubilizes odor substances, cleans sensory receptors, and traps xenobiotics [8]. The olfactory region is mainly supplied by a branch of the olfactory artery.

### 2.1. Nose-To-Brain Delivery Pathways

The unique connection between the nasal cavity and the central nervous system makes the nasal mucosa a potential adsorption site for central nervous system therapeutics with minimal invasiveness, avoiding the passage via the blood–brain barrier and the gastrointestinal and first pass degradation, thus improving drug bioavailability.

Currently, the exact mechanisms responsible for the nose-to-brain delivery of drugs are not well known; however, several evidences suggest that the trigeminal, olfactory, and systemic pathways are the main driving forces in delivering drugs via the nose-to-brain route.

Presumably, although all these mechanisms contribute to the transport of drugs to the central nervous system, the properties of the drug and the formulation used may determine the prevalence of one of these pathways [31].

#### 2.1.1. The Olfactory Pathway

The olfactory pathway has been long considered to be the main pathway for the nose-to-brain delivery of drugs. In fact, the olfactory region contains olfactory neural cells that provide a direct portal between the nose and the central nervous system, terminating as olfactory receptors into the mucus layer and projecting into the olfactory bulb. Moreover, olfactory neural cells undergo a continuous turnover, making the nasal barrier an open environment due to the constant rearrangement of tight junctions [32,33].

In order to be delivered to the central nervous system following the olfactory pathway, drugs have to cross the olfactory epithelium. This process can occur via two main mechanisms: extracellular and intracellular [34].

The extracellular pathway involves the external transport of drugs into the perineural space, after reaching the lamina propria, and depending on the chemical properties of the drug, it includes paracellular and transcellular mechanisms.

The transcellular pathway is responsible for the transport of lipophilic drugs across the olfactory epithelium and occurs mainly through sustentacular cells via endocytosis (receptor mediated or fluid phase) or passive diffusion, depending on the lipophilicity of the drug.

On the other hand, hydrophilic drugs cross the olfactory epithelium mainly via paracellular diffusion through tight junctions and/or clefts surrounding sustentacular and olfactory neural cells. This process is slower than the transcellular pathway, and it is strongly influenced by the molecular weight of the drug and the formulation [19].

Once they have crossed the olfactory epithelium and reached the lamina propria, drugs are transported to the central nervous system extracellularly through the perineural space, within olfactory ensheathing cells and olfactory nerve fibroblasts, by bulk flow mechanisms [35,36]; in this regard, probably, the conformational changes that also occur during the impulse propagation through the olfactory neural cells axons may contribute to the transport of drugs into the perineural space [37].

On the other hand, the intracellular mechanism involves the internalization of the drug via endocytosis or pinocytosis into olfactory neural cells, followed by the intracellular axonal transport into the nervous central system.

Olfactory neural cells have shown the ability to internalize some pathogens, such as herpes and polio viruses, large biological molecules, such as enzymes (i.e., horseradish peroxidase) and proteins (i.e., albumin and insulin), as well as gold nanoparticles, aluminum salts, and so on [38,39,40,41].

Pinocytosis is considered the main mechanism of drugs internalization into olfactory neural cells due to the high specificity of endocytosis [24]. Once inside, endosomes are translocated along axons toward the olfactory bulb where the drug is released via exocytosis and distributed in other central nervous system sites thanks to the projections existing between the olfactory bulb and other brain regions, such as the olfactory tract, hypothalamus, amygdala, etc. [34,42].

Unlike the extracellular pathway, the intracellular mechanism is a really slow and inefficient process, requiring 24 h to reach the central nervous system [39,43].

Moreover, the neuronal uptake is a quite specific process not applicable for a broad spectrum of drugs. In light of this, the extracellular pathway seems to be the predominant mechanism of transport, since several published studies highlight a rapid delivery and distribution of intranasally administrated drugs into the central nervous system [44,45,46].

#### 2.1.2. The Trigeminal Pathway

The trigeminal pathway involves the branches of the trigeminal nerve that innervate the olfactory and respiratory region.

The trigeminal nerve is the largest of the twelve cranial nerves and can be divided into ophthalmic, maxillary, and mandibular branches that convey sensory information from the nasal cavity, ocular mucosa, and oral cavity to the central nervous system. In particular, the ophthalmic and maxillary divisions are mainly involved in the nose-to-brain route, since they innervate the nasal mucosa while the mandibular branch innervates mainly the oral cavity.

The trigeminal branches present two different central nervous system entry points at the level of the pons and the olfactory bulb, respectively [5,18,47,48].

As already reported for the olfactory pathway, the delivery of drugs to the central nervous system following the trigeminal route can occur via extracellular pathways or intra-axonal transport.

The intra-axonal transport, via the trigeminal route, has been observed for several intranasally administrated agents such as lidocaine, insulin-like growth factor-I, interferon β, vascular endothelial grow factor, nanoparticles, etc. [49,50,51,52,53,54].

In addition to the trigeminal and olfactory pathways, it may be that other nerves, i.e., the facial nerve, contribute to the nose-to-brain delivery of drugs [18].

## 3. Nose-To-Brain Delivery: Limitations

Besides the great potential of the intranasal route for central nervous system therapeutics, there are however several disadvantages and problems related to both the administration site, the drug, and pharmaceutical formulation used (Figure 2).

Firstly, the small surface area of the nasal cavity severely limits the volume of the formulation that can administered, thus reducing the concentration of the adsorbed drug. For this reason, only low-dose active drugs and small agents can be effectively administered via this route in order to prevent to alter the physiological nasal functions [7,55]. Additionally, since the nasal cavity is covered by a thick layer of lipophilic mucus and the transcellular and intracellular pathways are thought to be mainly culprits for the nose-to-brain delivery, the intranasal route seems to be more suitable for lipophilic drugs; in fact, it has been estimated that the bioavailability is extremely low for hydrophilic molecules (10%) and peptides (<1%) [56].

Another factor that strongly limits drug adsorption at the level of the nasal mucosa is the mucociliary clearance. It is a defense mechanism performed by the combined action of mucus and ciliated cells that prevents the entrance of xenobiotics into the respiratory system transporting them to the nasopharynx where they are expectorated or ingested, and, consequently, it negatively affects drug retention time into the nasal cavity; in fact, it has been estimated that nasal formulations have a half-life time of about 20 min [57]. Herein, the use of mucoadhesive agents, such as chitosan, cellulose derivatives, and polyacrylates can limit the mucociliary clearance and increase the retention time [58].

In addition to the mucociliary clearance, drug absorption and bioavailability can be negatively affected by the peptidase enzymes that are present on the nasal epithelium via enzymatic degradation [59].

However, besides the physiological and structural aspects of the nasal mucosa and cavity, drug absorption is also strongly influenced by several factors correlated to the drug and the pharmaceutical formulation itself. In fact, one of the main drawbacks associated with the intranasal administration is the irritation of the nasal mucosa induced by the drug itself and/or the pharmaceutical formulation due to some parameters such as pH and osmolarity and excipients like cosolvents, permeation enhancers, polymeric, and metallic nanoparticles [60,61]. Moreover, it has to be considered that some drugs could potentially have harmful effects on the olfactory and trigeminal nerves [62].

## 4. Stimuli-Responsive Hydrogels

Hydrogels are three-dimensional networks formed by crosslinked hydrophilic polymers capable of absorbing large amounts of water. Recently, they have been studied as potential delivery platforms for biomedical applications due to their biocompatibility, biodegradability, nonimmunogenicity, and tunable properties [63].

Hydrogels can be divided into physical or chemical gels depending on the type of crosslinking that can be based on covalent bonds or weak physical interactions. In particular, physical crosslinking shows the advantage of obtaining hydrogels with tunable properties and responsivity to different external cues such as pH, temperature, and ionic modulation [64]. In this regard, in order to overcome the limitations of the intranasal route, stimuli-responsive hydrogels have emerged as an interesting strategy for the intranasal administration of drugs thanks to their ability to increase drug retention time and bioavailability and decrease mucociliary clearance [65]; in fact, they can be administered as liquid formulation, guaranteeing an accurate administration, and then, after instillation, they undergo sol–gel transition triggered by external physiological factors such as pH, temperature, ionic modulation, etc., leading to an increase of drug retention time and protection from enzymatic degradation [7].

Several mucoadhesive polymers such as chitosan, pluronic, carbopol, and cellulose derivatives exhibit sol–gel behavior in response to several cues and, consequently, can be used for stimuli-responsive hydrogels preparation in order to further improve retention time and drug absorption [66].

Moreover, the use of hydrogels as drug delivery systems shows the advantage to obtain a high drug-loading efficiency and sustained release, thus reducing the dose frequency and improving patient compliance and, thanks to their viscosity, a prolonged retention time [67].

Stimuli-responsive hydrogels can be divided in several classes depending on the external cues, and among them, thermos-, pH-, and ion-responsive gels are the more efficient delivery platforms for intranasal administration [68].

Thermoresponsive gels show sol–gel transition in response to a specific temperature change. Ideally, the temperature range should be at physiological values around 25–37 °C in order to guarantee an easy and accurate administration and avoid early drug loss [7].

Among thermoresponsive polymers, poloxamer, in particular poloxamer 407 and 188, is the most frequently used gelling agent for the preparation of in situ gels thanks to its mucoadhesiveness and sol–gel transition at physiological temperature values. It is a triblock polymer composed of one unit of polyoxypropylene and two units of polyoxyethylene that undergo sol–gel transition driven by supramolecular entanglements of micelle consequent to the dehydration of the polyoxypropylene units [69]. However, poloxomer-based hydrogels show low mechanical strength and viscosity in physiological conditions that limit their use, and, consequently, in order to overcome this problem, usually poloxamer is mixed with other polymers such as carbopol, chitosan, cellulose derivatives, etc. [70,71,72].

In this context, Ahmed et al. designed a poloxamer-/chitosan-based thermoresponsive gel loaded with agomelantine and evaluated its suitability and efficacy for the delivery of antidepressant drugs via the nose-to-brain route [73]. Firstly, the hydrogel was prepared by physically mixing chitosan (0.5% *w*/*v*), an agomelantine-based nanoemulsion and poloxamer 407 (20% *w*/*v*) (Figure 3). Chitosan was added in order to increase the mucoadhesiveness of the formulation; in fact, the calculated mucoadhesive strengths were 3747.75 dynes/cm^2^ and 6246.27 dynes/cm^2^ for the formulation without and with chitosan, respectively, and, moreover, the obtained hydrogel showed high strength and viscosity, advantageous features that further enhance drug retention time and adsorption. In fact, the pharmacokinetics studies highlighted a ~three-fold higher drug bioavailability in the brain, when administered intranasally compared to the intravenous administration of the drug solution.

In addition to poloxamer, other polymers such as chitosan usually mixed with polyols, cellulose derivatives like ethyl hydroxy-ethyl cellulose, and xyloglucan are used as gelling agents for the preparation of thermoresponsive hydrogels [74].

In this context, Dalvi et al. recently reported the development of thermosensitive xyloglucan-based hydrogels and evaluated their potential use as pharmaceutical formulations for the nose-to-brain delivery of rufinamide, an antiepileptic drug marketed as tablets and oral suspension that, however, shows low oral bioavailability and brain distribution [75]. In order to enhance rufinamide bioavailability, thus improving also patient compliance, in this study, tamarind seed xyloglucan thermosensitive derivatives were obtained by partial removal of galactose residues via enzymatic degradation and used for in situ gels preparation via the cold method. The formulation containing 2.0% *w*/*v* of xyloglucan derivatives was evaluated as more suitable for intranasal administration in terms of sol–gel transition temperature, viscosity, strength, release profile, and mucociliary inhibition. Promising results emerged from in vivo pharmacokinetic studies highlighting a greater brain distribution for the in situ gel in terms of brain targeting indices such as direct targeting efficiency percentage and brain drug-direct-transport percentage with an AUC_0→tlast_ around two-fold higher compared to the drug suspension, always administrated intranasally, thus proving to be a promising alternative formulation for rufinamide administration.

On the other hand, pH-responsive gels are systems that show sol–gel transition triggered by pH changes. Carbopol, a polyacrylate derivative, is a pH-sensitive polymer showing sol–gel transition at physiological pH and, consequently, largely used for the preparation of pH-sensitive gels for biomedical applications. Regarding the intranasal administration, the slightly acidic pH of the nasal cavity ionizes the carboxylic groups of the polymer, and this consequently leads to gel formation due to the electrostatic repulsions with the nasal mucosa [76]. However, compared to the thermoresponsive systems, pH-sensitive gels are much less studied for intranasal administration.

In this regard, a noteworthy example of intranasal pH-sensitive hydrogel has been proposed by Sherje et al. In this study, hydrogels based on carbopol 934 and hydroxypropyl methyl cellulose were prepared and loaded with a hydroxypropyl-β-cyclodextrin and paliperidone inclusion complex and evaluated for the intranasal administration [77]. The resulting systems showed high mucoadhesiveness and viscosity, sol–gel transition at adequate pH values (5.1–5.3), biocompatibility, controlled drug release, and, moreover, in vitro permeation studies highlighted a great drug nasal permeation attributable to the presence of hydroxypropyl-β-cyclodextrin that acts as an adsorption enhancer.

Lastly, ion-responsive gels are systems that undergo sol–gel transition induced by ions. Among ion-sensitive polymers, gellan gum, an anionic polysaccharide, is one of the most used gelling agents for the preparation of ion-sensitive hydrogels since it shows sol–gel transition in contact with physiological fluids (i.e., nasal) induced by the cationic complexation especially with Ca^2+^ [66,78].

In this regard, interestingly, Jelkmann et al. recently developed new gellan gum amino derivatives and evaluated their suitability for intranasal administration [79]. The amino derivates were prepared via a two-step synthesis involving first an oxidative cleavage of diol groups followed by a reductive amidation, obtaining polymers with different degree of amination. Subsequently, the synthetized derivatives were used for the preparation of hydrogels that were subsequently characterized in terms of viscosity, mucoadhesiveness, and mucociliary clearance inhibition. The obtained hydrogels showed enhanced viscosity and mucoadhesiveness compared to the gellan gum ones, thanks to the amino groups that interact with the nasal mucus via ionic interactions, and moreover, reversible inhibition of ciliary movement and absence of cytotoxicity were observed, thus making these platforms potential formulations for the nose-to-brain delivery of drugs.

This section reviews the recent advances made in the last years on the design of stimuli-responsive hydrogels for the nose-to-brain delivery of central nervous system therapeutics for the treatment of neurodegenerative diseases such as Alzheimer’s disease and Parkinson’s disease, depression, schizophrenia, and brain stroke.

### 4.1. Stimuli-Responsive Hydrogel for the Treatment of Alzheimer’s Disease

Alzheimer’s disease is the most common form of dementia, determining a progressive cognitive decline, involving first memory and subsequently other cognitive domains such as visuospatial, language, and thinking functions [80]. Physiopathologically, it characterized by the presence of β -amyloid plaques and neurofibrillary tangles of hyperphosphorylated forms of tau protein in the brain [81]. Moreover, several evidences suggest that also neuroinflammation and microglial activation play a key role in the pathogenesis of Alzheimer’s disease [82]. At the moment, unfortunately, there are no approved anti-Alzheimer effective drugs associated with a repair or regeneration of the neural tissue and/or able to block the neurodegenerative process, but rather current therapeutic approaches act mainly on reducing symptoms and slowing the neurodegeneration and include the oral administration of acetylcholinesterase inhibitors such as donezepil, rivastigmine, tacrine, galantamine, and NDMA receptors antagonists such as memantine. However, these drugs show low bioavailability at the brain level due to the BBB and gastrointestinal and first-pass degradation.

In order to overcome this problem and increase drug bioavailability, an interesting strategy has been recently proposed by Abouhussein et al. who developed thermoresponsive hydrogels for the nose-to-brain delivery of rivastigmine [83]. Several hydrogels were prepared using two different poloxamer types (407 and 188) and four mucoadhesive agents such as chitosan, hydroxypropyl methyl cellulose, carbopol 934, and sodium carboxy methyl cellulose via cold method, and their rheological behavior, mechanical strength, mucoadhesiveness, and release profile were evaluated. The formulation based on poloxamer 407 and carbopol 934 showed the best combination in terms of sol–gel transition temperature, mechanical properties, and mucoadhesiveness and, consequently, ex vivo permeation and in vivo pharmacokinetics studies were performed on it. Ex vivo results highlighted a much higher nasal permeation of rivastigmine incorporated into the gel compared to the drug solution (84% vs. 28%, respectively), attributable to the mucoadhesiveness of pluronic and carbopol that enhance drug retention time and decrease mucociliary clearance. Moreover, pharmacokinetic studies showed a greater rivastigmine cerebral distribution for the stimuli-responsive hydrogel compared to the intranasal and intravenous administration of the drug solution, with C_max_ (%/g) values of 0.58, 0.2, and 0.16, respectively, and a much higher bioavailability with AUC (%min/g) values of 84.70, 14.65, and 16.86, respectively, making this platform a promising formulation for the intranasal administration of anti-Alzheimer drugs (Figure 4).

In addition to the conventional anti-Alzheimer drugs, since neuroinflammation and microglial activation are thought to play a key role in the pathogenesis of Alzheimer’s disease, some natural antioxidant compounds such as curcumin, resveratrol, and saponins have emerged as potential anti-Alzheimer agents. However, these agents show low bioavailability and cerebral distribution, thus limiting their use and efficacy [84,85,86].

In this regard, recently, Chen et al. designed a thermo- and pH-sensitive hydrogel for the nose-to-brain delivery of timosaponin BII and evaluated its potential use and efficacy for the prevention of Alzheimer’s disease [87]. The gelling agents poloxamer 407 and gellan gum were used for the preparation of the hydrogel via the cold method in order to obtain a dual-sensitive (thermo and pH) system. In vivo studies were performed on murine models of neuroinflammation and amyloidosis induced by lipopolysaccharide and highlighted that timosaponin BII improved memory and language functions, partially reducing the cognitive decline induced by LPS and inhibited iNOS expression with consequent reduction of the levels of some proinflammatory mediators such as TNF-α, IL-1β, and, therefore, neuroinflammation.

### 4.2. Stimuli-Responsive Hydrogels for the Treatment of Parkinson’s Disease

Parkinson’s disease is the second most common neurological disease characterized by bradykinesia, rigidity, and rest tremor, thus leading to a progressive motorial function decline. Physiopathologically, it is characterized by a progressive neuronal degradation in the substantia nigra, and, consequently, shortage of dopamine and accumulation of Lewy’s body at the brain level [88,89].

Current available treatments include levodopa, dopaminergic agonists such as ropinirole and pramipexole, monoamine oxidase-B inhibitors such as selegiline and rasagiline, anticholinergic agents, and amantadine [90].

Due to the motorial function decline induced by Parkinson’s disease, dysphagia can often occur in patients thus making difficult the oral administration of drugs, most common route for anti-Parkinson agents [91]. Moreover, several anti-Parkinson drugs show low oral bioavailability, such as rasagiline, a monoamine oxidase-B inhibitor with oral bioavailability of around 35% in humans, due to high gastrointestinal and first-pass degradation [92].

In this regard, in order to overcome these problems and increase rasagiline bioavailability and patient compliance, Ravi et al. designed thermosensitive gels and evaluated their potential use as intranasal formulations for rasagiline nose-to-brain delivery [93]. Several in situ gels were prepared by using two different poloxamer (407 and 188) types and two mucoadhesive agents such as carbopol 934P and chitosan via cold method, and their rheological behavior, drug release profile, effect on mucociliary clearance, and mucoadheseviness were evaluated. The optimal intranasal formulations resulted to be P407 15% *w*/*v*, P188 15% *w*/*v* with 0.3% *w*/*v* chitosan or Carbopol 934P, showing sol–gel transition temperature at physiological values (30–34 °C), sustained drug release within 4 h, great mucoadhesiveness, with no significative difference between the two different mucoadhesive polymers and no nasal toxicity. Pharmacokinetics studies were performed and showed a six-fold and four-fold higher bioavailability for rasagiline incorporated into poloxamer-/chitosan- and poloxamer-/Carbopol-based gels, respectively, compared to the oral solution, and, furthermore, higher C_max_ values(~5.3 ug/mL and 7.8 ug/mL for intranasal in situ formulations with carbopol and chitosan, respectively, and ~2.5 ug/mL for the oral solution), making this delivery platform a promising formulation for the nose-to-brain delivery of central nervous system drugs (Figure 5).

Among dopaminergic agonists used as anti-Parkinson agents, we find rotigotine, marketed as a transdermal patch called Neuropro due to a low oral bioavailability consequent to a massive hepatic degradation. However, this transdermal formulation showed some crystallization problems that lead to its withdrawal by FDA [94].

In this regard, interestingly, in order to develop a proper formulation for the nose-to-brain delivery of rotigotine, Wang et al. designed delivery platforms based on a thermosensitive gel incorporating rotigotine-loaded polymeric micelles [95]. Several thermosensitive formulations were prepared by using two types of Poloxamer (407 and 188) in different ratios via physical mixing into aqueous solution with rotigotine-loaded polymeric micelles obtained via solvent evaporation method. The optimal intranasal formulation resulted to be 22% *w*/*v* P407 and 2% *w*/*v* P188, showing sol–gel transition temperature at physiological values ~32 °C, sustained drug release within 48 h, and no nasal toxicity. Promising results emerged from in vivo pharmacokinetics studies highlighting an extensive and fast brain distribution of rotigotine, about 2.5–3.5-fold higher in the olfactory bulb, cerebrum, cerebellum, and striatum compared to intravenous administration.

### 4.3. Stimuli-Responsive Hydrogels for the Treatment of Depression

Major depressive disorder is one of the most common psychiatric condition characterized by altered brain functions such as memory, attention, feelings, ability to think and concentrate, fatigue, sleep, and eating disorders, thus negatively impacting patients’ quality of life and representing one of the most common causes of disability [96].

Its etiology is complex and multifactorial, and, currently, the exact mechanisms responsible for its pathogenesis are not clearly known. However, several models have been proposed for major depressive disorder etiology: neuroinflammation, by reducing the levels of monoamines and increasing the levels of tryptophan catabolites, overactivity of the hypothalamic-pituitary-adrenal axis, and reduction of neurogenesis and neuroplasticity [97,98].

At present, the main pharmacological treatments act primarily on the monoaminergic system by increasing the synaptic levels of monoamines such as serotonin, norepinephrine, and dopamine and include selective serotonin reuptake inhibitors, that act mainly on the serotonin transporter (SERT) such as fluoxetine, paroxetine, and citalopram, serotonin-norepinephrine reuptake inhibitors, that act on both SERT and norepinephrine transporter (NET) such as duloxetine and venlafaxine, and tricyclic antidepressants such as imipramine and amitriptyline [99]. Among them, the selective serotonin reuptake inhibitors represent the most used antidepressants class thanks to their higher tolerability compared to the others; however, unfortunately, most of them show low oral bioavailability due to an extensive first-pass degradation [100]. In this regard, Thakkar et al. developed intranasal formulations for paroxetine nose-to-brain delivery [101]. Several in situ gels were prepared by using gellan gum as gelling agent, low-molecular-weight hydroxypropyl methyl cellulose as mucoadhesive agent, and hydroxypropyl-β-cyclodextrin as solubilizer and permeation enhancer, and their viscosity, strength, mucoadhesiveness, and loading efficiency were evaluated. The optimal intranasal formulation resulted to be 0.3% *w*/*v* gellan gum, 0.10% *w*/*v* hydroxypropyl methyl cellulose, and 10% *w*/*v* hydroxypropyl-β-cyclodextrin, showing adequate stiffness, strength, viscosity and mucoadhesive force, excellent loading efficacy (~93%), sustained drug release profile with ~80% of the drug released within 6 h, ease of spraying, and no nasal toxicity. Moreover, promising results emerged from in vivo pharmacokinetics studies highlighting an extensive and fast brain distribution of paroxetine after intranasal administration of the in situ gel compared to the oral administration of the drug suspension with T_max_ values of 0.5 h and 4 h, C_max_ (ng/mL) values of ~870 and 590, respectively, making this delivery platform a promising formulation for the nose-to-brain delivery of central nervous system drugs.

A noteworthy further example of intranasal stimuli-responsive hydrogel for the nose-to-brain delivery of antidepressant drugs has been reported by Avachat et al. [102].

In this study, a thermoresponsive gel was formulated by using poloxamer 407 as a gelling agent and PVP K30 as a mucoadhesive agent and evaluated for the nose-to-brain delivery of venlafaxine. The optimized formulation showed gelation temperature at physiological values ~32 °C, adequate viscosity, strength and mucoadhesive force, and no nasal toxicity. Pharmacodynamic studies were performed on rats and showed a better efficacy as antidepressant for venlafaxine on the basis of forced swim and locomotive activity tests after the intranasal administration of the thermoresponsive hydrogel compared to the oral solution.

### 4.4. Stimuli-Responsive Hydrogels for the Treatment of Schizophrenia

Schizophrenia is a psychiatric disorder characterized by altered brain function such as memory, feelings and mental processing, hallucinations, delusions, and social withdrawal [103]. Its etiology is multifactorial, but genetic and environmental factors seem to have a key role [104]. Currently, the mechanisms responsible for the pathophysiology of schizophrenia remain still unclear; however, strong evidences suggest that the dopaminergic and the glutamic neurotransmission may be involved in the development of negative and positive symptoms, respectively.

The main pharmacological treatments act mainly on the dopaminergic circuit by blocking the D2 receptor and include clozapine, risperidone, haloperidol, paliperidone, etc. Antipsychotic drugs are mainly administrated orally and primarily used in acute episodes of schizophrenia alleviating positive symptoms [105].

In addition to conventional treatments, in the last years, a new class of antipsychotic peptide drugs is emerging as a promising therapeutic approach against schizophrenia showing a higher safety profile with reduced side effects compared to the conventional therapies. However, unfortunately, one of the main problems associated with this class of antipsychotics is related to their route of administration, since they cannot be administered orally due to an extensive enzymatic degradation. In this context, the intranasal route turns out to be a potential administration site thanks to its noninvasiveness, possibility of the nose-to-brain delivery, and avoidance of GI degradation; in this regard, a noteworthy example of antipsychotic peptide drug nose-to-brain delivery system has been recently proposed by Majcher et al. [106]. In this work, an innovative intranasal formulation based on oxidized starch nanoparticles and carboxymethyl chitosan has been designed and evaluated for the delivery of a preclinical antipsychotic peptide that acts as allosteric modulator of D2 receptor. In particular, promising data came from in vivo studies that highlighted a full alleviation of schizophrenia-related negative symptoms for up to 3 days after the intranasal administration of the carboxymethyl chitosan and starch nanoparticles-based gel compared to the drug solution alone, administrated nasally or intraperitoneally, indicating that the intranasal formulation increased drug nasal permeation and cerebral bioavailability and efficacy.

### 4.5. Stimuli-Responsive Hydrogels for the Treatment of Brain Injury

Stroke is the second cause of death worldwide and one of the major causes of disability. It is caused by impaired perfusion and blockage of cerebral blood vessels that lead to hypoxia and, consequently, cell death and cerebral damage. Stroke can be ischemic or hemorrhagic: ischemic stroke is caused by blood vessels occlusion and usually results in thrombotic and embolic conditions, while hemorrhagic stroke is caused by blood vessels rupture, and it is associated with higher mortality.

At present, therapeutic approaches are limited and involve mainly reperfusion treatments such as intravenous thrombolysis [107]. However, there are many natural and synthetic agents that have gained attention due to their neuroprotective effect; in this regard, Xie et al. developed an intranasal formulation combining polyamidoamine dendrimers with an in situ gellan-bum-based gel for the nose-to-brain delivery of paenol, a neuroprotective agent that was found to reduce cerebral stroke in murine models. Firstly, polyamidoamine dendrimers were prepared and loaded with paenol obtaining an encapsulation efficiency of ~54%, and, subsequently, incorporated in an ion-sensitive gel obtained by using gellan gum 0.45% *w*/*v* and hydroxypropyl cellulose 0.3% *w*/*v* via cold method. The developed formulation showed advantageous features such as low critical ion concentration, facilitating the sol–gel transition under physiological condition, adequate viscosity and strength guaranteeing easy of spraying, and sustained release profile with ~80% of the drug released within 12 h. Moreover, fluorescence studies investigating the nose-to-brain delivery mechanism were performed and highlighted a greater cerebral bioavailability of paenol after the intranasal administration of the gel compared to the drug solution [108].

## 5. Conclusions

In the last years, neurological diseases have resulted in a global health issue, representing the first cause of disability and the second cause of death worldwide.

However, unfortunately, the currently available treatments against neurological diseases are not associated with a repair and/or regeneration of the neural tissue and, consequently, resolution of the clinical situation, but primarily act on alleviating symptoms and slowing the neurodegenerative process. The main limitation of central nervous system therapeutics is related to their delivery to the nervous system in therapeutic quantities due to the presence of the BBB that prevents the entrance of xenobiotics into the nervous systems. In order to overcome these limitations, the intranasal route has recently emerged as a promising administration site for central nervous system therapeutics providing a direct connection between the nose and the central nervous system via the trigeminal and olfactory pathways.

Despite the great potential of the intranasal route for central nervous system therapeutics, there are however several disadvantages related to the anatomy and physiology of the administration site such as the small surface area of the nasal cavity, mucociliary clearance, and enzymatic degradation that lead to a reduction of drug retention time and, consequently, absorption. In order to overcome these limitations, stimuli-responsive hydrogels have emerged as promising delivery systems for the nose-to-brain delivery of central nervous system therapeutics. Several mucoadhesive polymers such as chitosan, pluronic, carbopol, and cellulose derivatives exhibit sol–gel behavior in response to several cues and, consequently, are widely used for stimuli-responsive hydrogels preparation, thus allowing one to obtain delivery systems with excellent mucoadhesinevess, viscosity, strength, and sustained and advantageous release profile, enhancing drug bioavailability and efficacy and patient compliance. Moreover, several studies regarding in-situ hydrogels for the nose-to-brain delivery of central nervous system drugs report great pharmacokinetic properties showing a higher brain distribution of the drug compared to the intravenous and intranasal administration of the drug solution.

For these reasons, in situ gels turn out to be promising pharmaceutical formulations for the intranasal administration of central system therapeutics. However, beside their great potential, they have never reached clinical studies, and only preclinical studies are reported. Thus, more detailed in vivo studies are clearly needed in order to better evaluate their efficacy and safety profile. Moreover, before reaching clinical studies, the exact mechanisms responsible for the nose-to-brain delivery of drugs should be better investigated and clarified. In fact, the translation of animal models to human clinical studies is complex due to anatomical and pathophysiological differences and has to be handled carefully finding precise and correct methods and models. In this regard, in addition, more detailed toxicological studies regarding gelling agents and excipients are clearly needed in order to better evaluate their safety and biocompatibility.

However, on the basis of the current data and findings, it is likely that in situ hydrogels will find a place into clinical practice for the nose-to-brain delivery of central nervous system drugs thanks to their biocompatibility, biodegradability, sustained release profiles, and ability to increase drug retention time, adsorption, and brain distribution.

## Figures and Tables

**Figure 1 materials-14-01802-f001:**
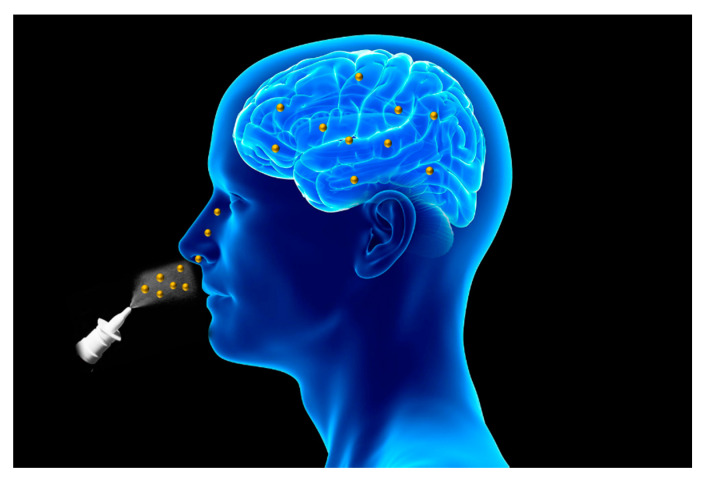
Schematic representation of intranasal route for drug delivery to the brain.

**Figure 2 materials-14-01802-f002:**
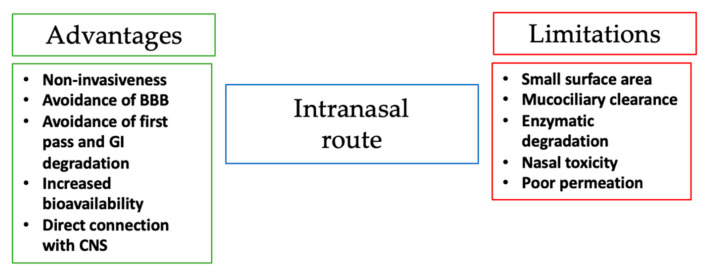
Advantages and limitations of the intranasal route.

**Figure 3 materials-14-01802-f003:**
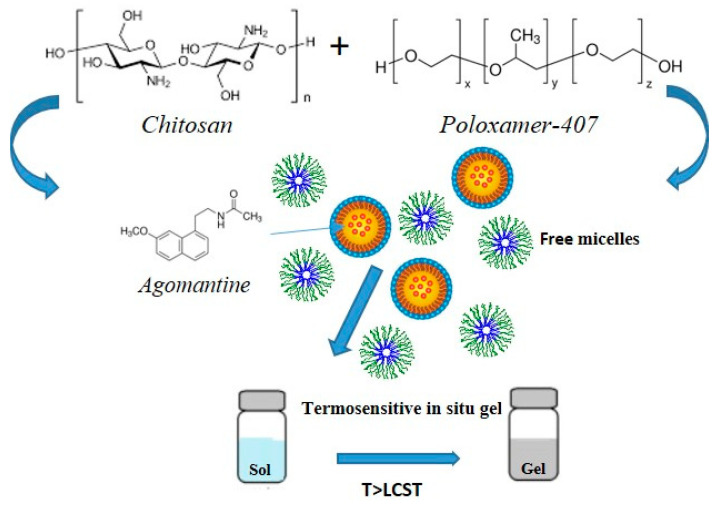
Schematic representation of gel preparation and sol–gel transition mechanism.

**Figure 4 materials-14-01802-f004:**
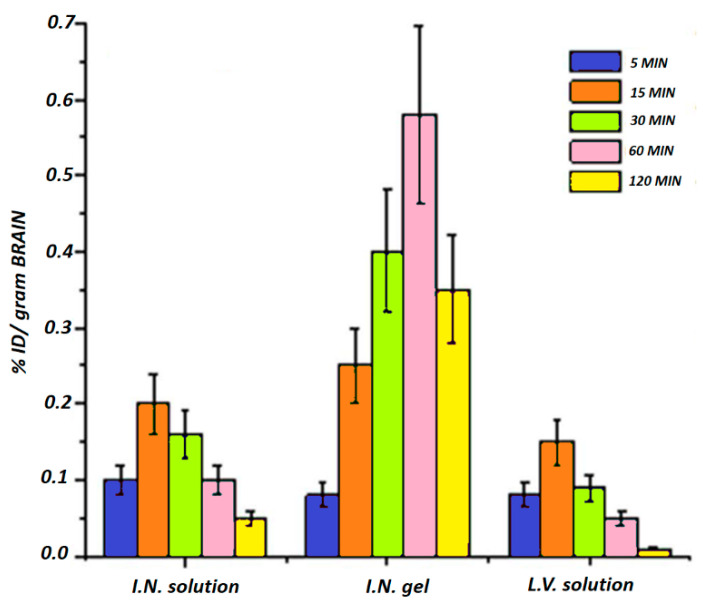
Distribution of rivastigmine in the brain after the intranasal administration of drug solution and rivastigmine-loaded pluronic-/carbopol-based gel and the intravenous administration of the drug solution. Adapted with modification from reference [83].

**Figure 5 materials-14-01802-f005:**
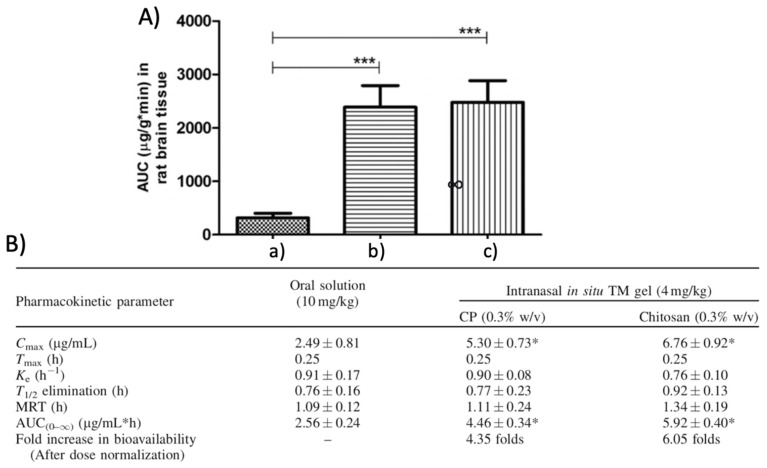
Cerebral distribution of rasagiline after intranasal administration of: drug solution (**a**), rasagiline-loaded Carbopol-based hydrogel (**b**) and rasagiline-loaded chitosan-based hydrogel (**c**) in rat. *** Indicates extremely significant difference in the compared values at *p* < 0.01 (**A**). Pharmacokinetic parameters of rasagiline for the oral solution and the intranasal carbopol- or chitosan-based in situ gel after administration in rats. C_max_: maximum plasm drug concentration, T_max_ time at which C_max_ is reached, K_e_: elimination rate costant, T_1/2_: time at which plasmatic drug concentration is decreased by 50%, MRS: Mean residence time and AUC_(0–∞)_: area under the curve. * Indicates that the values are significantly different from oral solution at *p* < 0.05 calculated using one way ANOVA followed by Dunnett’s multiple comparison test. (**B**). Adapted with permission form Taylor and Francis [93].

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
