# Peer review of "Biomaterials for Drugs Nose–Brain Transport: A New Therapeutic Approach for Neurological Diseases"

_materials, 2021, doi:10.3390/ma14071802_

Round 1

Reviewer 1 Report

The authors summarize the recent advances in intranasal drug delivery materials for neurological diseases. This is an interesting topic to be discussed.
The review has been well written. However, some defects exist and need improvements.

The authors only discussed stimuli-responsive hydrogel for treatment of AD and PD. Applications of hydrogel to treat other neurological diseases, including brain ischemia and other neurodegenerative diseases, should be discussed as well.

The section "2. nose to brain delivery: limitations" should be arranged as "4. nose to brain delivery: limitations" before the conclusion section.

some minor defects:

1, The "Figure 2. Schematic representation of intranasal route for drug delivery to the brain" should be "Figure 1. Schematic representation of intranasal route for drug delivery to the brain".

2, The "2.1.1. The trigeminal pathway" should be "2.1.2. The trigeminal pathway"

3, The "2. Nose to brain delivery: limitations" should be "3. Nose to brain delivery: limitations", if this section is not removed to the back of the reivew.

Author Response

Reviewer 1

The authors summarize the recent advances in intranasal drug delivery materials for neurological diseases. This is an interesting topic to be discussed.

The review has been well written. However, some defects exist and need improvements.

The authors only discussed stimuli-responsive hydrogel for treatment of AD and PD. Applications of hydrogel to treat other neurological diseases, including brain ischemia and other neurodegenerative diseases, should be discussed as well.

As the reviewer suggested, applications of stimuli-responsive hydrogels for the treatment of other neurological diseases such as schizophrenia, depression and brain stroke have been discussed and the following paragraphs have been added:

4.3 Stimuli-responsive hydrogel for the treatment of depression(line 509-555)

4.4 Stimuli responsive hydrogels for the treatment of schizophrenia(line 556-585)

4.5 Stimuli responsive hydrogels for the treatment of brain stroke(line 586-607)

The section "2. nose to brain delivery: limitations" should be arranged as "4. nose to brain delivery: limitations" before the conclusion section.

As the reviewer suggested in minor defects, the section has been changed to 3. We decided to put this section before the section of stimuli-responsive hydrogels, since these platforms represent a promising approach to overcome intranasal route limitations.

some minor defects:

1, The "Figure 2. Schematic representation of intranasal route for drug delivery to the brain" should be "Figure 1. Schematic representation of intranasal route for drug delivery to the brain".

As the reviewer requested, the mistake has been corrected.

2, The "2.1.1. The trigeminal pathway" should be "2.1.2. The trigeminal pathway"

As the reviewer suggested in minor defects, the subsection has been arranged as 2.1.2.

3, The "2. Nose to brain delivery: limitations" should be "3. Nose to brain delivery: limitations", if this section is not removed to the back of the reivew.

As the reviewer suggested in minor defects, the section has been arranged as 3.

Reviewer 2 Report

" Biomaterials for drugs “nose-brain” transport: a new therapeutic approach for neurological diseases" - there seems to be a typo in the title.

This a review on an important topic that looks to review the current state and future perspectives on nose-to-brain delivery for using novel biomaterials.

I have some comments below:

1. Unfortunately, at present, there are no fully effective treatments against neurodegenerative diseases, since they are not associated with a regeneration of the neural tissue, but rather act on slowing the neurodegenerative process.

The above sentence is not clear - need to rephrase it.

Also, there are several other grammatical errors - need to re-check.

2. Figure 2. 

Schematic representation of intranasal route for drug delivery to the brain - need to be improved - in the current state is hard to understand.

3. The authors need to summarise the limitations in a table or image to provide a snapshot view.

4. Conclusion section needs to provide clarity about what needs to be done next in terms of pre-clinical and clinical research and future perspective in this area.

Author Response

Reviewer 2

 " Biomaterials for drugs “nose-brain” transport: a new therapeutic approach for neurological diseases" - there seems to be a typo in the title.

As the reviewer suggested, this mistake has been corrected.

This a review on an important topic that looks to review the current state and future perspectives on nose-to-brain delivery for using novel biomaterials.

I have some comments below:

  1. Unfortunately, at present, there are no fully effective treatments against neurodegenerative diseases, since they are not associated with a regeneration of the neural tissue, but rather act on slowing the neurodegenerative process.

The above sentence is not clear - need to rephrase it.

As the reviewer suggested, this sentence has been rephrased as follows:

“However, unfortunately, the currently available treatments are not associated with a repair and/or regeneration of the neural tissue and, consequently, resolution of the clinical situation, but primarily act on alleviating symptoms and slowing the neurodegenerative process”

Also, there are several other grammatical errors - need to re-check.

As the reviewer suggested, the review has been rechecked and mistakes corrected.

  1. Figure 2.

Schematic representation of intranasal route for drug delivery to the brain - need to be improved - in the current state is hard to understand.

As requested, a new figure was inserted.

  1. The authors need to summarise the limitations in a table or image to provide a snapshot view.

As the reviewer suggested a new figure(Figure 2) summarizing limitations and advantages of the intranasal route has been added(line 260).

  1.  Conclusion section needs to provide clarity about what needs to be done next in terms of pre-clinical and clinical research and future perspective in this area.

According to reviewer’s suggestion the following sentences have been modified and added to the conclusion section:

“However, beside their great potential, they have never reached clinical studies and only preclinical studies are reported. Thus, more detailed in vivo studies are clearly needed in order to better evaluate their efficacy and safety profile. Moreover, before reaching clinical studies, the exact mechanisms responsible of the nose to brain delivery of drugs should be better investigated and clarified. In fact, the translation of animal models to human clinical studies is complex due to anatomical and pathophysiological differences and has to be handled carefully finding precise and correct methods and models. In this regard, also, more detailed toxicological studies regarding gelling agents and excipients are clearly needed in order to better evaluate their safety and biocompatibility.”

Round 2

Reviewer 2 Report

I am satisfied with the revision and would commend the authors for their efforts.